# Virtual Reality Utilized for Safety Skills Training for Autistic Individuals: A Review

**DOI:** 10.3390/bs14020082

**Published:** 2024-01-23

**Authors:** Lili Liu, Xinyu Yao, Jingying Chen, Kun Zhang, Leyuan Liu, Guangshuai Wang, Yutao Ling

**Affiliations:** 1National Engineering Research Center of Educational Big Data, Central China Normal University, Wuhan 430079, China; liulili@mail.ccnu.edu.cn (L.L.); chenjy@mail.ccnu.edu.cn (J.C.); zhk@mail.ccnu.edu.cn (K.Z.); lyliu@mail.ccnu.edu.cn (L.L.); 2National Engineering Research Center for E-Learning, Central China Normal University, Wuhan 430079, China; 3Faculty of Artificial Intelligence in Education, Central China Normal University, Wuhan 430079, China; yaoxinyu031962@mails.ccnu.edu.cn; 4College of Physical Science and Technology, Central China Normal University, Wuhan 430079, China

**Keywords:** virtual reality, autism spectrum disorder, safety skills, skills training

## Abstract

In recent years, virtual reality technology, which is able to simulate real-life environments, has been widely used in the field of intervention for individuals with autism and has demonstrated distinct advantages. This review aimed to evaluate the impact of virtual reality technology on safety skills intervention for individuals with autism. After searching and screening three databases, a total of 20 pertinent articles were included. There were six articles dedicated to the VR training of street-crossing skills for individuals with autism, nine articles focusing on the training of driving skills for individuals with ASD, and three studies examining the training of bus riding for individuals with ASD. Furthermore, there were two studies on the training of air travel skills for individuals with ASD. First, we found that training in some complex skills (e.g., driving skills) should be selected for older, high-functioning individuals with ASD, to determine their capacity to participate in the training using scales or questionnaires before the intervention; VR devices with higher levels of immersion are not suitable for younger individuals with ASD. Second, VR is effective in training safety skills for ASD, but there is not enough evidence to determine the relationship between the level of VR immersion and intervention effects. Although the degree of virtual reality involvement has an impact on the ability of ASD to be generalized to the real world, it is important to ensure that future virtual reality settings are realistic and lifelike. Again, adaptive models that provide personalized training to individuals with ASD in VR environments are very promising, and future research should continue in this direction. This paper also discusses the limitations of these studies, as well as potential future research directions.

## 1. Introduction

Autism spectrum disorder (ASD) is a pervasive developmental disorder that manifests itself in early childhood and often persists throughout life. It is characterized by language delays, impairments in social and communication functioning, narrowed interests, attention deficits, and stereotyped behaviors [1]. In March 2023, the Centers for Disease Control and Prevention (CDC) released the latest autism prevalence screening results, which showed that 1 in 36 children had autism spectrum disorder, or about 2.8%, with increase of 0.5% from the previous year [2]. The CDC has reported that the leading cause of death in children aged 1 to 14 is unintentional injury, and the population most vulnerable to accidents is children with special needs. The individual characteristics and deficits of individuals with ASD may increase their risk of injury [3] or death [4]. Therefore, it is important to incorporate safety skills training into treatment plans for individuals with ASD.

An important factor to be considered for skills training is whether the training takes place in a man-made or natural environment [5]. The majority of conventional intervention methods, such as peer-regulated interventions, video modeling, social storytelling, and cognitive behavioral therapy (CBT), are conducted in natural settings and rely on people converting their knowledge into visual information so that particular skills can be acquired and applied to real-world situations. However, for certain safety skills, training in natural settings might be difficult because the training environment might be dangerous (e.g., crossing a street) or unethical (e.g., dealing with cuts or avoiding fires). As a result, there is a growing need for intervention methods that both ensure safety and are more effective.

In recent years, an increasing number of researchers have begun to use virtual reality (VR) as an assistive therapy tool for individuals with ASD in the fields of socialization, communication, daily living, and cognitive skills [6,7]. VR is an equally suitable candidate for training in safety skills. Virtual environments (VEs) offer a compromise between artificial and natural environments, with clear instructional goals and designs that meet the standards of teaching and learning environments for people with autism. They also offer controlled and individualized conditions that allow individuals with ASD to train in a controlled, repeatable environment without safety concerns [8]. Simultaneously, it offers diverse immersive experiences for users, including non-immersive, semi-immersive, and immersive [9]. Burdea and Coiffet summarize immersion, interactivity, and imagination as the basic characteristics of VR technology [10], also known as the 3I characteristics. Although the effectiveness of VR interventions has been measured and assessed in different ways across studies, in general, VR interventions are effective if the skills learned in the VR environment can be generalized to the natural environment. In addition, the use of VR in autism interventions has multiple theoretical supports: firstly, according to psychological theories of autism, the cognitive style of individuals with ASD is predominantly visually oriented [11]. VR technology emphasizes specifically audio–visual responses. Notably, they prefer controlled interactions and are able to respond well to challenges provided by computers with clear structure and consistent expectations. Moreover, learning theories such as task-based learning, adaptive learning, situated learning, and simulation learning further enhance the theoretical support for the use of VR in ASD intervention, making its application even more convincing. However, there are some problems with intervention methods using VR. First, VR interventions have to be combined with real life, and skills acquired in VEs need to be tested in the real world and further trained and developed, which has received less investigation. Second, exposure to some immersive VR devices may cause individuals to experience discomfort similar to motion sickness, eyestrain, headache, nausea, and sweating. Thirdly, nowadays, VR resources and activities for personalized interventions for the ASD population are still scarce.

Despite these challenges, VR intervention is still a promising approach. Numerous researchers have shown that VR technology is useful in the field of ASD intervention. For instance, Smith et al. developed a VR job interview training system (VR-JIT), and, after the training, the interviewing abilities of people with ASD dramatically improved [12]. A VR social cognitive intervention system (VR-SCT) created by Kandalaft et al. improved the social cognition, social functioning, and social skills of autistic people [13]. Kim et al. discovered that after participating in a VR social interaction program, children with ASD showed a greater understanding of facial emotions [14]. ASD language skills improved following training, according to Nubia et al.’s design of an application that may teach word and semantic knowledge to children with ASD [15]. Numerous previous studies have focused on the social communication, emotion recognition, language, and cognitive aspects of people with ASD, but it is also essential to conduct research on the implementation of particular skill training (driving skills, street-crossing skills, etc.) for people with ASD to promote independence and safety.

Crossing the road, flying, taking public transportation, and driving are mobile scenarios that many people must face in their daily lives, and these skills are directly related to an individual’s independence and participation in society. It is often difficult for individuals with ASD to develop these adaptive skills needed to achieve independence in real-world settings, so VR may be a practical training medium. Therefore, this study provides a systematic review of research on the use of VR to train these four categories of safety skills. In this review, we provide suggestions for the optimal intervention age, intervention devices, and personalized interventions for each skill. The following three issues are discussed in this study:What is the appropriate age for ASD patients to apply VR for safety skills training?Are VR treatments effective, and does the level of VR immersion affect the effectiveness of the intervention and its effect on generalization?How can we use VR technology to implement personalized safety skills training for people with autism?

## 2. Materials and Methods

### 2.1. Inclusion Criteria and Exclusion Criteria

The study had to be published in a peer-reviewed article.The study had to include at least one patient diagnosed by a clinician or identified as having ASD using a standardized diagnostic tool.The study had to use virtual reality for ASD intervention or training to provide effectiveness or feasibility results.The study had to implement skill-specific interventions that improved the independence and safety of people with ASD.The intervention means had to be VR devices with different levels of immersion (immersive, semi-immersive, non-immersive), and the intervention content had to include a variety of safety skills.Studies that did not include a therapeutic intervention as the independent variable were excluded.

### 2.2. Research Process

For a systematic search, the following databases were selected by the researchers: PubMed, Web of Science, and ScienceDirect. The search was conducted in March 2023 and was restricted to articles published in the past 10 years (2013–2023). Search keywords were as follows: “autism OR Autism Spectrum Disorder OR ASD OR autism*” and “Virtual Reality OR VR OR immersive technology* OR semi-immersive technology* OR non-immersive technology* “and “train OR intervention”. The search was filtered by the English language, with peer-reviewed articles only. The conference literature was excluded from the review after discussion among the authors. A total of 1438 articles were retrieved from the initial search. The search protocol was repeated by the second author to enhance the reliability of the search, and the same number of papers was identified. After removing duplicates, a total of 910 articles were obtained. A Microsoft Excel spreadsheet and the Endnote documentation management tool were used in this process.

After the database search, an initial screening of the literature was completed based on the title and abstract, and papers that did not include VR and ASD were initially excluded. When the information provided by the title and abstract was insufficient for exclusion, we retained articles first. After discussion and consensus, 852 articles were removed and 58 were included. A comprehensive review of these 58 articles was conducted in the second round of screening. After screening the articles based on the inclusion and exclusion criteria, 38 articles were excluded, and a total of 20 articles were added to a Microsoft Excel spreadsheet for coding, coding each article in terms of the following variables: (1) the number of topics, (2) background information about the experimental subjects (gender, age, etc.), (3) the VR device utilized, (4) the level of immersion in VR, (5) the intervention goals, (6) the maintenance and generalization of the learned skills, (7) the number and duration of the interventions. Two authors participated in the review and coding process, and, if there was disagreement between the coders during the screening process, to achieve a consistent outcome, they consulted one another and sought an expert’s opinion before making a decision. The literature screening process is shown below (Figure 1).

## 3. Results

### 3.1. Intervention Purpose

One of the most significant and instrumental activities of daily living for integration into society and into the community is the ability to move. Training in this capability can enhance the independence and safety of people with ASD in their daily lives. This review considers four categories of safety skills, street-crossing skills, driving skills, air travel skills, and bus-riding skills, and provides a summary of the relevant literature over the past decade. Overall, 30% (n = 6) of these 20 studies used VR to train people with ASD in street-crossing skills [16,17,18,19,20,21]. Street crossing is one of the daily life skills that everyone has to master and may require more effort and time to train for individuals with ASD. VR can provide an effective training environment, allowing them to train in a controlled environment without safety concerns. Fornasari et al. trained individuals with ASD in urban virtual environments to navigate and explore the area [21]; Dixon et al. taught children with ASD how to recognize safe road-crossing conditions in an immersive VR environment [19]. In addition to street-crossing skills, driving skills are also needed to foster individual independence, and 45% (n = 9) of studies focused on training driving skills for people with ASD [22,23,24,25,26,27,28,29,30], e.g., embedding an adaptive model within a driving simulator in VR, adjusting the task difficulty through the acquisition and analysis of multimodal data from the user’s driving task, and the personalization of the driving task to achieve the optimal training effect [23,24]; helping ASD individuals to reduce their fear of driving and improve their driving cognition and performance [29]. Meanwhile, 10% (n = 2) of the studies focused on training ASDs in air travel skills [31,32], e.g., Miller et al. conducted a training program for ASD patients using an iPhone X and Google Cardboard device that lasted 3 consecutive weeks, followed by a real air travel rehearsal at San Diego International Airport in the fourth week [32]. Three (15%) of the studies were dedicated to improving the public transportation skills of individuals with ASD [33,34,35]. Table 1 below provides a summary of each study.

### 3.2. Intervention Subjects

Of the 20 studies included, most specified the characteristics of the experimental subjects and inclusionary or exclusionary criteria, excluding individuals with ASD with severely impaired vision, those who were unable to recognize the color of traffic lights and street signs, and those who engaged in aggressive behaviors [17,18]; they also excluded individuals with a history of seizures or co-occurring serious medical conditions, such as epilepsy, cancer [21,31], intellectual disabilities or mental deficits, brain injuries, and those with genetic disorders or chromosomal anomalies [22]. In addition, some studies proposed subject selection criteria: the use of verbal language as the primary mode of communication [35]; possessing necessary skills, such as following instructions, maintaining attention, and so on; no visual or auditory impairments [19]; a reading ability of 4th grade or above; fluency in English, etc. [29]. Most studies included subjects with high-functioning autism.

The number of subjects in the included studies ranged from a minimum of 3 to a maximum of 51, with over half of the studies selecting more than 10 subjects in the age range of 4 to 44 years old. Four research studies focused on younger age groups (4–12 years old), while two others chose older persons with ASD (19–44 years old) for their intervention, in studies on teaching street-crossing abilities to autistic patients. The age range of participants in the driving skills training study for patients with ASD was 13 to 29. In two studies of air travel skills training for ASD, the age range of participants in one study was 4 to 8 years old, and the age range of participants in the other study was 10 to 22 years old. The age range of subjects in the studies on bus riding with ASD was 18–34 years old. The age range of subjects selected for training in the same skill varied considerably from study to study.

In the literature selected for this study, 374 patients with ASD participated in the studies, with a majority of males. Female ASD patients were not included in the studies in some of the literature. The ASD male-to-female ratio of 3:1 could account for the fact that there was a gender gap in the patients’ male-to-female ratio [36], even though the researchers did not restrict themselves to recruiting only men. To ensure that the results are applicable to female patients, future studies should take the gender balance into consideration when choosing their subjects.

### 3.3. Application of Virtual Reality

VEs can be presented in a variety of ways, ranging from traditional computer interface-based video games to full-body interactive systems involving motion capture and avatars. VR can provide users with various types of immersive experiences: immersive (e.g., head-mounted displays; VR CAVE), semi-immersive (e.g., large screen displays), and non-immersive (e.g., desktop displays) [37].

#### 3.3.1. Application of Immersive Virtual Reality

Six (35%) of the 20 selected studies used immersive VR environments. Modern immersive VR environments are typically created by display devices based on wrap-around screen projections or head-mounted displays (HMDs). Display devices based on wrap-around screen projection, such as the VR CAVE, typically consist of multiple projection walls and motion trackers that track the posture of the user’s head and body, and these devices enable a group of users to share a virtual experience within the same physical world. For example, Tzanavari et al. used VR CAVE to conduct a four-day training program for six children with Pervasive Development Disorder—Not Otherwise Specified (PDD-NOS), to teach them how to cross the road safely. Results indicated that most children were able to achieve the desired goal of learning the task, which was verified by having them cross a real pedestrian crossing [16]. HMDs range from the relatively low-cost Google Cardboard to more advanced HMDs such as the Oculus Rift, Oculus Quest, and HTC VIVE. In a study on air travel skills for individuals with ASD using a Google Cardboard device on the iPhone X, the effectiveness of the intervention was later verified at a real-world airport [32]. Schmidt et al. used the Google Cardboard and Google Daydream HMD to train people with ASD in bus-riding skills, after which they enacted what was learned in the real world [33,35]. For an immersive VR environment consisting of an Oculus Rift headset and sensors designed to train individuals with ASD in street-crossing skills and bus-riding skills, the results indicated that all participants achieved mastery in both the VR and natural environment settings [19,34]. Although these devices allow for more immersive interactions, they can lead to excessive sensory stimulation and uncomfortable symptoms such as dizziness, headaches, and nausea, which may have a greater impact on people with ASD, who tend to have higher levels of perceptual awareness and sensitivity to stimuli.

#### 3.3.2. Application of Semi-Immersive Virtual Reality

Using an interactive device such as the Microsoft Kinect in combination with a projector can create a semi-immersive VR environment that provides a more intuitive and interactive way to learn, while mitigating sensory overstimulation. Five (25%) studies used semi-immersive VR. For example, Saiano et al. tested the effectiveness of using a combination of a markerless motion capture device (Microsoft Kinect) and virtual reality to teach skills such as street crossing and following street-sign directions to individuals with autism. The study outcomes indicated the feasibility of interacting with a VE through a natural interface to facilitate the acquisition of street safety skills among adults with ASD [17]. Cox et al. designed a VR driving simulation training (VRDST) pod that interacted with the virtual environment through a driving console. Participants were assessed pre- and post-training for driving-specific executive function (EF) abilities and tactical driving skills. This study demonstrated the feasibility and potential efficacy of VRDST for novice ASD drivers [22].

#### 3.3.3. Application of Non-Immersive Virtual Reality

In addition, common VR interactive devices include keyboards, mice, joysticks, and touchscreens, on which non-immersive VR environments can be created. Five studies (25%) used non-immersive virtual environments. Saiano et al. used a gamepad (GP) interface to interact with VR to train people with autism to learn crossover and tracking skills [17]. Fornasari et al. designed a desktop virtual environment to train ASD individuals to navigate around the city and interact with VE using a mouse [21]. Both studies demonstrate the effectiveness of VR interventions. Using a desktop VR minimizes motion sickness for ASD individuals in VE, and the desktop VR interaction device is inexpensive and easy to promote.

To accommodate the heterogeneity of individuals with autism, one study suggested selecting appropriate VR devices to interact with based on the characteristics of individuals with autism. The study by Tan et al. designed a serious game to train individuals with autism in road-crossing skills, which can be operated on a variety of platforms, such as computers, iPads, tablets, or cell phones, and the user can interact with the game through different electronic input/output devices, including Microsoft Kinect motion sensors, mice, keyboards, and touchscreens [20]. In conclusion, immersive, semi-immersive, and non-immersive virtual environments have their own advantages and disadvantages, and different studies should select appropriate VR devices based on the purpose of the intervention and the characteristics of individuals with ASD.

### 3.4. Research Methods

Of the 20 papers included, five used comparative studies, two of which were on individuals with ASD and individuals with typical development (TD) [21,34], and one compared the effects of different VR devices on interventions for individuals with ASD [17]. Two studies compared the effectiveness of interventions in VR versus traditional settings [22,26]. Sixteen studies used a single-subject design. VR interventions mostly lasted between 3 and 10 weeks. The shortest intervention lasted only 5 days, with children with ASD being trained in a VR CAVE environment for the first 4 days on how to safely cross a crosswalk, and then taken to a real street on the fifth day to validate the training [16]. The longest intervention lasted up to three months and included 12 interventions and an evaluation after three months [22]. Three articles did not mention the total intervention duration, and eight articles did not specify the duration of each intervention. Most studies ranged from 20 to 40 min for one intervention to 45 to 60 min or even 90 min for longer interventions. 

In summary, the majority of the articles in our research utilized a single-subject design, while only a few articles used a multiple-probe design or multiple-baseline design. Each of these three research methods has its advantages and disadvantages. Firstly, single-subject designs allow for the detailed and precise analysis of individual behavior changes. This is particularly useful when studying unique cases or individual differences and is relatively easy to implement. However, the findings may not be easily generalized to a larger population and the validity of the findings may be questioned. Secondly, multi-probe designs allow researchers to study the effects of an intervention on multiple behaviors simultaneously. Compared to the traditional single-subject design, this method can be more time-efficient. The drawback, however, is that changes in one behavior may influence or confound the results of another, and analyzing and interpreting data from multiple behaviors can be more complex than focusing on a single behavior. Finally, the design provides a strong demonstration of experimental control by showing that changes in the dependent variable coincide with the introduction of the independent variable. However, this research method can be time-consuming. In summary, the choice of research method depends on the specific research question, the nature of the behavior being studied, and practical constraints.

Most studies described positive effects of the VR intervention. There are two ways to evaluate the effectiveness of a VR intervention: subjective and objective measurements. Subjective measures include questionnaires, (e.g., for individuals with ASD or for parents, therapists), interviews, and scales, etc. Objective measurements include eye-gaze data, electrodermal data, test scores, etc. The intervention duration for people with ASD should not be too long or too short; an overly long duration will make them bored and sleepy, and an overly short duration will not be conducive to the long-term maintenance of skills. Generally, 20–40 min is the ideal duration of intervention. Moreover, we suggest that future studies should employ more rigorous experimental designs to investigate the effectiveness of VR-enabled ASD interventions. 

### 3.5. Skill Generalization

Four studies out of 20 referred to the generalization of intervention skills. Of these, three studies provided the opportunity to test their generalization in the real world. Tzanavari et al. tested the ability to generalize the learned skills on a real crosswalk on the fifth day after a 4-day intervention with six children with ASD [16], and Dixon et al. tested the ability to generalize the learned skills in an experiment by having a clinician hold the subject’s arm while standing next to a real street and asking the subject, “Is it safe to cross the street?” The subjects formed their own judgments, without any prompting or feedback from the physician [19]. People with ASD practiced their skills in a real airport in the work by Miller et al. [32]. In one study, parents or guardians were asked to complete a questionnaire to assess the extent to which subjects improved their street-crossing behavior in real-world scenarios [18]. Overall, the results of generalized measures of skills were positive. The timing of the implementation of the generalization measure also varied across studies. Some studies assessed it immediately, one day after the end of the intervention, and some assessed the generalization a week or several months later. In conclusion, current studies lack the assessment of skill generalization for individuals with ASD, probably due to safety concerns and because different research takes different approaches to assessing generalization.

## 4. Discussion

The major goal of this review was to evaluate the effectiveness of VR technology in safety skills training for patients with ASD. A total of 20 research papers (published between 2013 and 2023 and obtained from the PubMed, ScienceDirect, and Web of Science databases) were included to analyze the empirical studies of VR technology for specific safety skills training in ASD, and the following three main questions were discussed.

### 4.1. Appropriate Age for Safety Skills Training Using VR in ASD Patients

One of the aims in this study was to analyze the optimal age for specific safety skills training for individuals with ASD. Most of the 11 studies that trained individuals with ASD in street-crossing skills, air travel skills, and bus-riding skills selected fewer than 10 subjects, which is a small sample size; for the same skill, there was a large variation in the age group of the subjects chosen for the different studies. For example, of the studies that trained people with ASD in street-crossing skills, four studies selected subjects in the lower age group (4–12 years old), and two studies selected a higher age range among adults with ASD (19–44 years old) for the intervention. Of the two studies of air travel skills training for ASD, one study selected subjects in the age range of 4–8 years old, and the other study selected subjects in the age range of 10–22. In addition, prior to the intervention, the researchers screened the subjects and selected only individuals with ASD who were capable of participating in the training; therefore, there was insufficient evidence to determine the optimal age range for the training of these skills.

Of the nine studies of driving skills training for ASD, more than 50% chose to have more than 20 subjects, and all of the VR interventions demonstrated positive effects. Driving skills are more challenging to develop for individuals with ASD because they are skills that require social communication, cognitive flexibility, and behavioral dexterity, and because individuals with ASD have difficulties in recognizing certain hazards associated with driving (e.g., traffic signals, road hazards, regulating speed, etc.) and deficits in attention shifting, continuous task performance, and visuomotor integration and coordination compared to their typically developing (TD) peers [38,39]. Individuals with ASD are also less likely than their peers to obtain a driver’s license, and even when they do acquire one, it occurs much later than for their peers [40]. Therefore, driving skills training for individuals with ASD should be selected for older, high-functioning patients.

In summary, it can be determined that training for these complex skills is not appropriate for low-functioning or younger individuals with ASD, because these skills place higher demands on the cognitive and behavioral abilities of the individual, and some of the skills require some rules to be taught before the intervention. For example, Saiano et al. had subjects practice the gesture vocabulary required to interact with a virtual environment (VE) prior to a street-crossing skill intervention [18]; Bian et al. had ASD users learn basic traffic rules and maneuvers before a driving skill intervention [30], which could be challenging for low-functioning or young people with ASD. In Saiano’s study, a subject was excluded from the training due to his inability to perform depth perception in the virtual environment. Therefore, when training some complex skills, it is important to determine the participants’ ability to participate in the training using questionnaires or scales before the intervention. 

While VR technology has shown positive results in the current research, we note that the HMD, an immersive VR device, is not suitable for younger ASD patients. Studies have shown that up to 80% of individuals will experience motion sickness symptoms such as dizziness, eyestrain, and nausea after using an HMD [41]. These uncomfortable symptoms can cause anxiety and mood swings in young children, which can be detrimental to the intervention. Mesa-Gresa et al. noted that studies of VR treatment for people with ASD have focused on the 8–14 age group [7]. Most VR companies offer recommended ages, such as the official Oculus document, which states that the product is not suitable for children under 13; the Samsung Gear VR requires users to be at least 13; and Sony’s PlayStation VR requires users to be at least 12. Therefore, a high-immersion VR CAVE, semi-immersive NI devices, or low-immersion desktop VR may be better options for young ASD patients. In summary, it is not true that the higher the level of immersion, the more effective it will be, and the selection of an appropriate VR intervention device should be based on the ASD individual’s own acceptance of the relevant device.

The purpose of training individuals with ASD in safety skills using VR equipment is to increase the safety and independence of individuals with ASD in their daily lives. High-functioning patients should be chosen because these abilities call for a high level of aptitude on the part of the person with ASD. In addition, for driving skills, individuals need to have a certain level of executive function (EF), and individuals’ EF usually reaches maturity at the age of 25 years, so driving skills training should be selected for older, high-functioning individuals with ASD. 

### 4.2. Effectiveness of VR Interventions and the Relationship between the Immersion Level of VR and the Intervention and Generalization Effects

Because VR technology is an emerging technology, its effectiveness for individuals with ASD remains uncertain. Determining whether VR is useful in enhancing safety skills for people with ASD was one of the objectives of this review. This research comprised a total of 20 empirical trials, and the overall findings suggested that VR had a favorable impact on ASD safety skills interventions. Although the final measurements varied, most studies demonstrated the effectiveness of VR interventions. Four studies used questionnaires; Tan et al. and Saiano et al. demonstrated that VR facilitated the acquisition of road-crossing skills and traffic sign recognition in ASD through a questionnaire assessment [17,20]. Miller et al. asked the parents of subjects to fill out a questionnaire to subjectively quantify their child’s air travel skill improvement, which showed that four out of five participants had improved scores (on a 5-point Likert scale) [31]; Baker-Ericze’n et al. came to the same conclusion after analyzing improved Driving Perceptions Questionnaire scores, which indicated that improvements in the anxiety levels, attitudes, and performance of ASD individuals occurred during driver training [29]. One study used an asynchronous multiple-baseline design to assess the effectiveness of the VR training of crossing skills in ASD, with low scores measured for each subject at baseline and progressively higher scores as the VR training progressed [19]. One study compared pre- and post-intervention measure scores and found substantial differences in each measure, indicating that the VR driving skill training (VRDST) system enhanced the driving abilities of people with ASD [22]. By analyzing physiological data from feedback, two trials demonstrated that the VR intervention decreased anxiety levels and boosted self-confidence during task performance. According to one study that employed a comparative experiment, ASD attitudes were more favorable in a VR training environment than they were in a traditional driving skills training setting [26]. Therefore, it is clear that VR is useful in teaching individuals with ASD safety skills. We found that most of the studies used subjective measurements, such as questionnaires, which, although more flexible, may be affected by subjective bias and the influence of personal stances and may lead to different conclusions in the same situation. Moreover, the results are difficult to quantify, which affects the persuasiveness of the experimental results. Therefore, we prefer to recommend the use of objective measures.

Regarding the relationship between the level of VR immersion and intervention effectiveness, only 1 out of 20 articles conducted a comparative study. Saiano et al. compared the effectiveness of a gamepad (GP, non-immersion)-based VR interaction mode and a natural interface (NI, semi-immersion)-based VR interaction mode in teaching ASD street-crossing skills. Natural interfaces, such as the Nintendo Wii and Microsoft Kinect, may mitigate the sensory overstimulation problem as they do not require sensors or devices in direct contact with the body of the user. The result suggested that the two interface types (NI and GP) were equally effective, and the learning performance was not affected by the level of VR immersion [17]. In addition, some studies found that the learning effect of ASD individuals in less immersive and fully immersive VR environments did not differ significantly [42]. Meanwhile, some individuals with ASD often refuse to participate in experiments because they cannot tolerate the excessive sensory stimulation brought about by immersive VR environments. According to an experimental study conducted by Malihi et al. that examined the variables influencing the VR experience of individuals with ASD, IQ and anxiety levels were both characteristics that had an impact on the ASD experience in VR environments [43]. As a result, there is no direct correlation between the degree of VR immersion and the effectiveness of the intervention. Instead, there may be a range of influences, including individual differences, the features of the intervention content, and the quality of the equipment used. Meanwhile, VR technology itself is still developing. The association between the degree of VR immersion and the intervention effect in many application situations and study fields still requires additional empirical studies. Future research needs to further explore the relationship between the level of VR immersion, intervention duration, and intervention effect to find the optimal intervention model for most individuals with ASD.

The ultimate goal of VR interventions is to generalize the learned skills to natural environments, and generalization is more effective if individuals are able to transfer the learned skills to real situations. However, there are still some challenges and limitations. (1) Environmental differences: VEs are often idealized, and it is difficult to fully simulate the various complexities of the real world. Real environments tend to face more uncertainty, change, and disruption, making it difficult to generalize. (2) Real-time feedback problem: learning via VR often allows for immediate feedback and guidance, but similar feedback may be difficult to obtain in real-life environments, and the feedback may not be sufficiently accurate and clear, which may have an impact on the transfer of skills. (3) Discomfort: some VR devices may cause discomfort to the users, affecting their learning experiences and learning outcomes, which in turn affects their generalization to real-life scenarios. Despite these limitations, VR remains a promising means of intervention.

Concerning the relationship between the level of VR immersion and generalization, we found that virtual environments with a higher degree of immersion were more conducive to generalization. This is because more immersive virtual environments do not have additional visual inputs and provide a more realistic experience for the user [44]. Miller and Bugnariu et al. argued that this realistic experience may increase the chances of the skills learned in VR environments being transferred to the real world [45]. Studies such as Saiano et al. have also shown that in comparison to the GP-based VR interaction mode, the NI-based VR intervention mode is more likely to help subjects to use the skills learned in the real world. Cobb et al. also mentioned that the more realistic the virtual environment is, the more generalization is possible [46], as the scenario is more “believable” and participants are more likely to connect and translate the learning and experience gained in the virtual environment to the real world. Therefore, future VR environments should be realistic and immersive.

In summary, the use of VR to train ASD individuals in safety skills is effective, but VR intervention effects are the results of multiple factors, and it is not true that the higher the level of VR immersion, the better the intervention. For training in safety skills, immersive VR is more appropriate for youth or adults with ASD, and immersive VR interventions are more conducive to generalizing the learned skills to the real world for individuals with ASD. However, if a person with ASD is unable to adapt to a more immersive VR, semi-immersive VR may be a good alternative that provides the user with a sufficient sense of experience while avoiding overstimulation. Finally, for autistic children, it is more suitable to perform training using desktop VR, but this tends not to generalize well. It may be more beneficial to use desktop virtual reality as a primary intervention medium before slowly transitioning to immersive virtual reality.

### 4.3. Using VR Technology to Provide Personalized Safety Skills Interventions for Individuals with Autism Spectrum Disorder

Personalized interventions are educational or therapeutic programs that are tailored to an individual’s characteristics and needs. Autistic people are in greater need of personalized intervention methods because of their heterogeneity. Traditional intervention methods pose high requirements for teachers, and all intervention sessions require guidance, with high costs and limited effects. Therefore, it is necessary to incorporate new technologies into VR environments to achieve personalized interventions.

Personalized interventions in VR environments are currently being used in relevant research. Out of the 20 publications included, six studies discussed how personalized training can be implemented in a VR environment. Bian et al., Baker-Ericze’n et al., and Wade et al. presented the development of a novel closed-loop adaptive VR driving simulator for individuals with ASD that can infer their engagement based on their physiological responses and adapt the driving task difficulty based on the engagement level in real time [25,29,30]. Zhang et al. developed and evaluated a visual information-based adaptive VR driving simulator to provide personalized driving interventions for users, and they conducted a small pilot study with 20 ASD adolescents using the system, which demonstrated the effectiveness of this adaptive driving system for intervention in ASD driving skills [24]. They also focused on electroencephalogram (EEG) and other physiological signals to construct an adaptive model that integrates multimodal information into the driving system. Multimodal information can produce better-tailored interventions and offers more accurate feedback than single-modal information.

Personalized interventions in VR environments are related to many factors, such as anxiety levels, the cognitive load, and so on, and it is important to consider which physiological indicators (e.g., blood volume pressure (BVP), galvanic skin response (GSR), skin temperature, eye-gaze data, heart rate, electrodermal data) can be used to determine the anxiety levels and cognitive loads of people with ASD in VR environments. Alkevicius et al. trained a model for anxiety level recognition using physiological signals such as BVP, GSR, and skin temperature [47]. A study by Rahman et al. also implemented a biofeedback framework to identify anxiety levels in terms of the heart rate and brain laterality index from acquired multimodal data [48]. The final results prove the validity and accuracy of these two models. These physiological-based adaptive models are independent of the intervention environment. Therefore, we can consider adding such models to existing VR serious games to enable personalized interventions for people with ASD. Moreover, we can continue to update it in the future to realize the upgrading and promotion of the models; with the development of wearable and less invasive physiological sensors, these models will be more reliable.

This adaptive model can adjust the task difficulty without human control, which is very useful for ASD personalized training and also helps to reduce the workload of therapists and improve the intervention effect. However, the existing research is still in the preliminary stage and numerous constraints need to be addressed in subsequent studies. First, it is challenging to obtain multimodal data on ASD patients in VR environments, and some of the data that are gathered are not useful. Second, data fusion approaches have their limits. Different data fusion methods may lead to different study results. Despite the above problems, the implementation of personalized interventions in VR environments is still a promising research direction. 

In order to adapt VR interventions to the various abilities and needs of ASD patients, in addition to considering the inclusion of such physiologically based adaptive models in VR environments, a comprehensive assessment of the cognitive, perceptual, and social abilities of ASD patients should be conducted before the use of VR, so as to facilitate the determination of the patients’ specific needs and level of adaptive ability. In addition, the VR intervention system should be adapted to different VR interaction devices. For instance, in order to adapt to the heterogeneity of different ASD individuals, Tan et al. developed a VR serious game that can support multiple platforms and interaction modes, such as Kinect sensors, touchscreens, mice, and keyboards [20]. At the same time, the setting of the VR environment should also fully consider the psychological characteristics and age characteristics of individuals with ASD. In conclusion, adding adaptive models to VR settings to offer individualized interventions for people with ASD is a promising research direction, and future researchers should continue to develop this field.

### 4.4. Limitations and Future Directions

In order to improve the independence and safety of individuals with ASD and to provide some guidance for future safety skills training, this study reviewed and analyzed the relevant literature of the past 10 years (2014–2023). Later, we will continue this study by including more literature year by year. Of course, this study is not without its limitations: the exclusion of published studies other than English language ones was among the limitations and challenges of this study. In addition, in terms of intervention effects, this paper focused on the influence of the degree of immersion in VR on it, and, in fact, the subjects’ age, intelligence, etc., may also represent influencing factors, which should be explored in future research.

## 5. Conclusions

This review summarized the research on VR for safety skills training in ASD over the past 10 years and included a total of 20 articles. It also discussed how individualized interventions for people with ASD can be carried out in VR environments, the best age group for safety skills training for ASD individuals in VR environments, the effectiveness of VR in safety skill interventions for ASD individuals, and whether the level of VR immersion affects the effectiveness of interventions and their generalization.

The results indicate that safety skills training for people with ASD using a virtual environment is effective, but these complex skills are not appropriate for training in younger or low-functioning people with ASD, so it is crucial to assess their capacity for participation in the training before the intervention using a scale or questionnaire. We advise choosing people with high-functioning ASD for training in bus, airplane, and street-crossing skills. Driving skills call for people to have more cognitive and behavioral flexibility, so older people with high-functioning ASD should be chosen for this type of training. In addition, younger ASD individuals may not be able to adapt to the excessive sensory stimulation induced by immersive virtual reality devices such as HMDs, and it is recommended that VR CAVEs, semi-immersive VR, or desktop devices be used to intervene in younger ASD individuals. Secondly, it is not true that the higher the degree of VR immersion, the better the intervention effect, Instead, overly immersive environments can cause users to experience motion sickness, diminishing the effectiveness of the intervention. Future virtual environments should be realistic and lifelike, to improve the transfer of newly acquired skills to the real world. Due to the heterogeneity of ASD patients and the large differences between individuals, personalized intervention is a worthwhile research direction. Existing research suggests integrating an adaptive model in the VR environment, which can be used to modify the difficulty of the task by analyzing the user’s physiological data, such as electrocorticography and eye movements. In the future, the development of the VR system should also take full consideration of the psychological and age characteristics of the individuals with ASD, and the content of the interventions should be adapted to the different VR devices, so that the appropriate intervention device can be selected based on the characteristics of the individuals with ASD.

Although these studies have had favorable results, there are still several flaws in the following areas. First, the method of measuring the skill performance of the subjects is relatively simple, and most studies use questionnaires to determine whether the subjects’ skills have improved, by having them answer questions or by asking their parents or therapists. Second, most of the experiments lack a post-experiment generalization assessment. Thirdly, the small sample sizes of some experiments result in less persuasive and credible experimental results. Finally, the majority of subjects selected in the experiments are high-functioning ASD individuals, and future research should also focus on safety skills training for low-functioning ASD patients.

## Figures and Tables

**Figure 1 behavsci-14-00082-f001:**
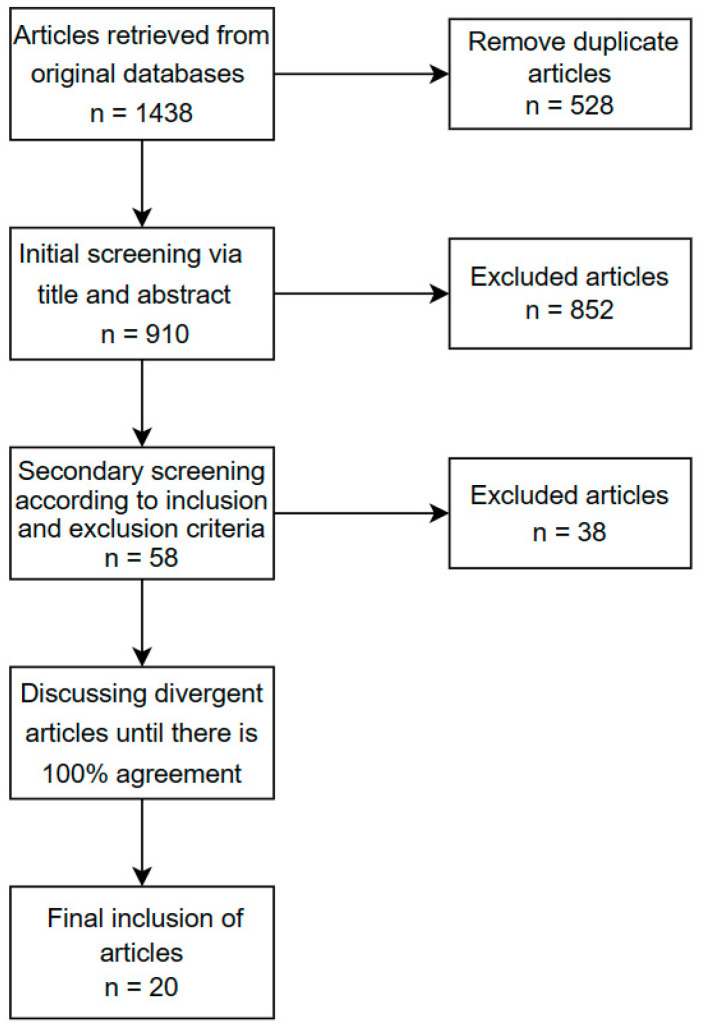
Flow chart of the study selection process.

**Table 1 behavsci-14-00082-t001:** Empirical research on use of VR to improve the safety skills of individuals with ASD.

Author	Sample Size	Age and Sex	VR Device	Level of Immersion	Intervention Purpose	Skill Generalization	Intervention Protocol
Fornasari, L., et al. [21]	16 ASD 16 TD	7–14	Mouse and computer	Non-immersive	Street-crossing skills		2 sessions, each of 45 min
Tzanavari, A., et al. [16]	6 ASD	8–11	VR CAVE	Immersive	Street-crossing skills	The experiment would end with a final session on a different day, at a real pedestrian crossing, to examine whether the children could generalize what they had learned.	8 sessions
Saiano, M., et al. [18]	7 ASD	9–44 all males	Natural interfaces	Semi-immersive	Street-crossing skills	Parents/legal guardians were also required to complete a questionnaire to assess to what extent they considered that subjects had improved their behavior in real-life situations.	10 sessions: familiarization (1–5) training (7–9) assessment (6 and 10) completion of questionnaire familiarization phase (30 min practice vocabulary of gestures) train phase (45 min)
Saiano, M., et al. [17]	10 ASD GP group (n = 6) NI group (n = 4)	NI group 19–44 GP group 19–31	Gamepad device and natural interfaces	Non-immersive and semi-immersive	Compare a modality of interaction with virtual environments based on the use of a classic gamepad with a modality based on a natural interface (Kinect) in the context of the acquisition of pedestrian skills	Parents/legal guardians were also required to complete a questionnaire to assess to what extent they considered that subjects had improved their behavior in real-life situations.	10 sessions, each session with a maximum duration of 45 min
Fan, J.F., et al. [23]	16 ASD	13–18 all males			Driving skill training		6 sessions, each of 60 min
Zhang, L., et al. [25]	20 ASD	13–18		Non-immersive	Real-time gaze-contingent driving Simulator capable of providing individualized feedback about how drivers scan their visual environment while driving		
Cox, D.J., et al. [22]	51 ASD	15.5–25 mostly males	A realistic driver’s cockpit with side and rear-view mirrors.	Non-immersive	Driving skill training	Driving-specific EF and general tactical assessments occurred at baseline and after 3 months of training.	14 sessions: 12 sessions for training and 2 assessments
Zhang, L., et al. [24]	20 ASD	13–18 mostly males		Non-immersive	Driving skill training		6 sessions, each of 60 min
Ross, V., et al. [26]		16–25		Semi-immersive	Driving skill training		8–12 sessions
Fan, J., et al. [27]	20 ASD	Mean age 15.29 years, mostly males			Driving skill training		6 sessions, each of 60 min
Simões, M., et al. [34]	10 ASD 10 TD	ASD Mean age 18.8 TD Mean age 21.9 years	Oculus Rift	Immersive	Teaching transportation skills		3 sessions, 20–40 min for each session
Dixon, D.R., et al. [19]	3 ASD	4, 6, 10 years old, mostly males	Oculus Rift headset and sensors	Immersive	Street-crossing skills		5 sessions, each session lasted 3–5 min
Bian, D., et al. [30]	23 ASD	Mean age 15.18, 21 males and 2 females	Computer	Non-immersive	Driving skill training		
BA, L.M., et al. [31]	5 ASD	4–8 4 males and one female	Smartphone and Google Cardboard	Immersive	Air travel skills	The fourth and final session was a real-world rehearsal	4 sessions
BA, L.M., et al. [32]	7 ASD	Mean age 18.28, 6 males and 1 female	iPhone X and Google Cardboard	Immersive	Air travel skills		3 sessions, 20 min for each session
Patrick, K.E., et al. [28]	48 ASD 48 TD	16–26			Driving skill training		
Schmidt, M., et al. [33]	5 ASD	22–34	Google Cardboard	Immersive	Teaching transportation skills	In the fourth stage, participants practiced skills in the real world with trained staff.	
Baker-Ericze’n, M.J., et al. [29]	19 ASD	15–29, mostly males			Driving skill training		8 sessions, 90 min for each session
Tan, Q.P., et al. [20]	5 ASD	5–12	Natural interface Kinect/keyboard/mouse/touchscreen	Semi-immersive	Street-crossing skills		
Schmidt, M., et al. [35]	6 ASD	Mean age 26.6, all males	Google Cardboard (HMD) and HTC Vive or Oculus Rift HMD	Immersive	Teaching transportation skills		3 sessions, 60 min for each session

Note: ASD: Autism Spectrum Disorder; TD: Typical Development; HMD: Head-Mounted Display.

## Data Availability

The data of this study are available from the corresponding author upon reasonable request.

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
