# Peer review of "Virtual Reality Utilized for Safety Skills Training for Autistic Individuals: A Review"

_behavsci, 2024, doi:10.3390/bs14020082_

Round 1

Reviewer 1 Report

Comments and Suggestions for Authors

This literature review addresses an extremely relevant and topical subject: the use of new technologies, and in particular virtual reality, to train and assess safety skills in people with autism spectrum disorders.
The authors have methodically followed a process of identification and selection of articles dealing with this subject over the decade. The specific questions concerning the interaction between individual characteristics (age of participants, diagnostic profile of participants, etc.) and characteristics linked to intervention modalities (nature of the virtual reality devices used, content and complexity of the intervention, duration of the intervention, etc.) are really very interesting and instructive, as are the perspectives proposed by the authors.

Some suggestions and comments are available in the attached file.

Comments on the Quality of English Language

Moderate editing of English language required to improve readers'understanding.

Author Response

Dear Reviewer:

Thank you very much for your comments and professional remarks concerning our manuscript entitled ‘Virtual Reality Utilized for Safety Skills Training for Autistic Individuals: A Review’ (ID: behavsci-2754222). These comments and suggestions are all valuable and very helpful for revising and improving our manuscript, and significantly important for our research. We have talked about these comments carefully and made modifications which we hope meet with your approval. We have provided a point-by-point response to your comments below.

Comment 1:  The authors make a very pertinent point here. Nevertheless, safety skills are broader than this and include, among other things, crossing the road, but also what to do in the event of fire, injury, poison ingestion, etc. The authors restrict safety skills to a particular category: those related to transport and travel. This is an important point, and should be made clearer by the authors.

Reply 1:  Thanks for your comments. As we know, one of the most significant instrumental activities of daily living for integrating into society and into the community is the ability to go out of home. Training in this capability can enhance the independence and safety of people with ASD in their daily lives. Although many events in the daily lives of people with ASD are associated with safety, crossing the road, flying, taking public transportation and driving are mobile scenarios that many people must face in their daily lives, and these skills are directly related to an individual's independence and participation in society. It is often difficult for individuals with ASD to develop these adaptive skills needed to achieve independence in real-world settings, so VR may be a practical training medium. No study has yet conducted an overview of such specific security skills. Therefore, this study has selected these four categories of safety skills for a review. However, we will further examine interventions for other safety skills in future studies.

Revised portions are marked blue in the manuscript (Page 2,3 Line 98-106; Page 4,Line 160-165).

Comment 2:  Could we have more details in the introduction about this particular point. The authors elaborate on it in the discussion, but perhaps at this stage readers could already have some explanation of the levels of immersion in virtual reality and the effectiveness of the interventions.

Reply 2:  Thanks for your comments. This is an excellent suggestion for us. We have already given a brief description of the levels of immersion in virtual reality and the effectiveness of the interventions in the introduction.

Revised portions are marked blue in the manuscript (Page 2, Line 63 - 69).

Comment 3:  This point deserves more attention: it is covered in section 2.2. Research process states that 852 items have been excluded, but the flow chart shows 452.

Reply 3:  We feel sorry for our carelessness. In our resubmitted manuscript, data in the flow chart has been corrected. Thank you for pointing this out.

Revised portions are marked blue in the manuscript (Page 4, Line 157).

Comment 4:  APA standards do not require the author's first name to be cited, but rather the abbreviated letters.

Reply 4:  Thank you for pointing this out. We have made changes using APA standards.  Revised portions are marked blue in the manuscript (Page 5,6, Table 1).

Comment 5:  This sentence is difficult to understand. A rewording might help readers.

Reply 5:  We have reworded this sentence. Modify as follows:

For example, Tzanavari et al(2015) used VR CAVE to conduct a four-day training for six children with PDD-NOS(Pervasive Development Disorder- Not Otherwise Specified)to teach them how to cross the road safely.

Revised portions are marked blue in the manuscript (Page 7, Line 229 - 231).

Comment 6:  ?

Reply 6:       We are really sorry for our careless mistakes. Thank you for your reminder.  Revise the sentence as follows:

Driving skills are more challenging for individuals with ASD because they are skills that requires social communication skills, cognitive flexibility, and behavioral dexterity.

Revised portions are marked blue in the manuscript (Page 10, Line 362).

Comment 7:  These difficulties could be clarified at this stage to facilitate reading.

Reply 7:  In the text we have added an explanation of these difficulties, as modified below:

         and because individuals with ASD have difficulty in recognizing certain hazards associated with driving (e.g., traffic signals, road hazards, regulating speed, etc.) and deficits in attention-shifting, continuous task performance, and visuomotor integration and coordination compared to their typically developing (TD) peers.

Revised portions are marked blue in the manuscript (Page 10, Line 363 - 365).

Comment 8:  We could use the terms children, teenagers or young people with ASD to designate individuals rather than the one used here.

Reply 8:  We think this is an excellent suggestion. We have replaced the original term with young people with ASD. Thank you for your advice.

Revised portions are marked blue in the manuscript (Page 10, Line 378).

Comment 9:  Would it be possible to have another brief explanation of the nature of this specific device?

Reply 9:  We have already given a brief description of the nature of this specific device in the text, as follows:

Natural Interfaces, such as Nintendo Wii and Microsoft Kinect, may mitigate the sensory over-stimulation problem as they do not require sensors or devices in direct contact with the body of the user.

Revised portions are marked blue in the manuscript (Page 12, Line 439 - 441).

Comment 10:  This sentence is extremely long and could be split in two to improve comprehension.

Reply 10:  Thanks to your suggestion, we've made new changes and improvements to this part. This part of the original text has been removed.

Revised portions are marked blue in the manuscript (Page 12, Line 441 - 444).

Comment 11:  To improve readers' understanding, it is essential to define the criteria for the effectiveness of Virtual environment interventions.

Reply 11:  We provide an explanation of the criteria for the effectiveness of VR interventions in the fourth paragraph of 4.2, as follows:

The ultimate goal of VR interventions is to generalize learned skills to natural environments, and generalization is more effective if individuals are able to transfer learned skills to real situations.

Revised portions are marked blue in the manuscript (Page 12, Line 459 - 461).

Comment 12:  Sugestion: virtual environment ?

Reply 12:   Thanks for your suggestions, we have made the following modifications:

The results indicate that safety skills training for people with ASD using a virtual environment is effective.

Revised portions are marked blue in the manuscript (Page 14, Line 572).

We tried our best to improve the manuscript and made some changes marked in blue in revised paper which will not influence the content and framework of the paper. We appreciate for your warm work earnestly, and hope the correction will meet with approval. Once again, thank you very much for your comments and suggestions.

Reviewer 2 Report

Comments and Suggestions for Authors

The article "Virtual Reality Utilized for Safety Skills Training for Autistic Individuals: A Review" presents an analysis of various studies on the application of virtual reality (VR) in training safety skills for individuals with Autism Spectrum Disorder (ASD). The paper reviews empirical studies, discussing the effectiveness of VR in this context, the relationship between the level of VR immersion and intervention outcomes, and the implementation of personalized safety skills training using VR technology. However, the paper needs to be revised following these suggestions and comments, before it could be considered for publication:

·         The article provides an extensive overview of relevant studies, but it could benefit from a more in-depth analysis of each study's methodologies and findings. This would help in better understanding the nuances and contextual factors affecting the outcomes of VR interventions in ASD.

·         The review lacks a critical evaluation of the methodologies used in the included studies. A more detailed examination of study designs, participant selection criteria, and intervention protocols would enhance the review's robustness.

·         While the paper discusses the generalization of skills learned via VR to real-world settings, it could more thoroughly address the challenges and limitations in achieving generalization. This is particularly crucial given the diversity in ASD symptomatology.

·         How do you define “VR technology”? Table 1 mentions mouse and computer, desktop as “VR technologies”.

·         The paper touches upon different VR technologies and their levels of immersion. However, a more detailed discussion on how different technologies specifically cater to or affect individuals with ASD would be beneficial.

·         While the paper acknowledges the need for personalized interventions, it could further elaborate on how VR interventions can be adapted to the wide range of abilities and needs within the ASD population.

·         The author could discuss related topics to autism such as public anxiety, see Enhancing biofeedback-driven self-guided virtual reality exposure therapy through arousal detection from multimodal data using machine learning. Brain Informatics (2023), Anxiety level recognition for virtual reality therapy system using physiological signals. Electronics (2019). Can the findings of this article be extended to related domains?

·         The findings discussed in Section 4 suggest that VR is effective in training safety skills but highlight that the level of VR immersion does not necessarily correlate with intervention effectiveness. It also emphasizes the need for personalized VR interventions due to the heterogeneity of ASD. While these findings are insightful, they are based on a limited number of studies and require further empirical validation to confirm the generalizability and effectiveness of VR interventions across different contexts and ASD populations.

·         The review could benefit from a more detailed analysis of the outcome measures used in the studies. Discussing the efficacy of VR interventions in a more nuanced way, considering different safety skills and individual differences, would be valuable.

Overall, while the article provides a comprehensive overview of the use of VR in safety skills training for individuals with ASD, a more detailed and critical analysis of the methodologies, technologies, and outcomes of the reviewed studies would strengthen its contribution to the field.

Author Response

Dear Reviewer:

Thank you very much for your comments and professional remarks concerning our manuscript entitled ‘Virtual Reality Utilized for Safety Skills Training for Autistic Individuals: A Review’ (ID: behavsci-2754222). These comments and suggestions are all valuable and very helpful for revising and improving our manuscript, and significantly important for our research. We have talked about these comments carefully and made modifications which we hope meet with your approval. We have provided a point-by-point response to your comments below.

Comment 1:  The article provides an extensive overview of relevant studies, but it could benefit from a more in-depth analysis of each study's methodologies and findings. This would help in better understanding the nuances and contextual factors affecting the outcomes of VR interventions in ASD.

Reply 1:  Thanks for your comments. We have provided a detailed description of the research methodology and findings of each study in sub-section 3.3 and 3.4.

Revised portion is marked red in the manuscript (Page 7,8, Line 222-267; Page 8, Line 280-284).

Comment 2:  The review lacks a critical evaluation of the methodologies used in the included studies. A more detailed examination of study designs, participant selection criteria, and intervention protocols would enhance the review's robustness.

Reply 2:  Thanks for your comments. All studies were included according to inclusion criteria. We add a critical evaluation of the research methodologies and study designs in sub-section 3.4 of the article. Participant selection criteria are described in detail in sub-section 3.2 and systematically analyzed at the end of this section. Because each study had a different intervention protocol, a systematic summary is not possible, and we provide a brief description of each study's research protocol Table 1 (Page 5, Line 184).

Revised portion is marked red in the manuscript (Page 9, Line 293 - 308; Page 7, Line 197、209 - 213).

Comment 3:  While the paper discusses the generalization of skills learned via VR to real-world settings, it could more thoroughly address the challenges and limitations in achieving generalization. This is particularly crucial given the diversity in ASD symptomatology.

Reply 3: Thanks for your comments, we agree with you very much. We add the challenges and limitations of generalizing the skills learned in virtual reality to real-world environments in the third paragraph of sub-section 4.2.

Added portion is marked red in the manuscript (Page 12, Line 457 - 468).

Comment 4:  How do you define “VR technology”? Table 1 mentions mouse and computer, desktop as “VR technologies”.

Reply 4: Thanks for your comments. We have replaced the term “VR technology” in Table 1 with “VR devices”.

Revised portion is marked red in the manuscript (Page 5, Line 184).

Comment 5:  The paper touches upon different VR technologies and their levels of immersion. However, a more detailed discussion on how different technologies specifically cater to or affect individuals with ASD would be beneficial.

Reply 5:  Thanks for your comments. We describe in the last paragraph of sub-section 4.2 on how different levels of immersion in virtual reality cater to and affect people with ASD.

Revised portion is marked red in the manuscript (Page 12, Line 484-492).

Comment 6:  While the paper acknowledges the need for personalized interventions, it could further elaborate on how VR interventions can be adapted to the wide range of abilities and needs within the ASD population.

  • The author could discuss related topics to autism such as public anxiety, see Enhancing biofeedback-driven self-guided virtual reality exposure therapy through arousal detection from multimodal data using machine learning. Brain Informatics (2023), Anxiety level recognition for virtual reality therapy system using physiological signals. Electronics (2019). Can the findings of this article be extended to related domains?

Reply 6: Thanks for your comments. We further elaborate in sub-section 4.3 on how virtual reality interventions can be adapted to suit the various abilities and needs of people with ASD, and refer to two pieces of literature you provided. What’s more, the findings of this paper can be extended to related domains.

Revised portion is marked red in the manuscript (Page 13, Line 504 - 552).

Comment 7: The findings discussed in Section 4 suggest that VR is effective in training safety skills but highlight that the level of VR immersion does not necessarily correlate with intervention effectiveness. It also emphasizes the need for personalized VR interventions due to the heterogeneity of ASD. While these findings are insightful, they are based on a limited number of studies and require further empirical validation to confirm the generalizability and effectiveness of VR interventions across different contexts and ASD populations.

Reply 7:  Thank you for your question, and we have added new references in Section 4 to support our point.

 First, skill interventions for individuals with autism via VR is still an emerging area of research, and most of the existing studies focus more on non-specific skills of autistic patients such as social communication, emotion recognition, and cognition. A number of previous reviews have demonstrated the effectiveness of virtual reality in training these non-specific skills. (see Virtual Reality Technology as an Educational and Intervention Tool for Children with Autism Spectrum Disorder: Current Perspectives and Future Directions. Behavioral Sciences (2022); Immersive Virtual Reality Enabled Interventions for Autism Spectrum Disorder: A Systematic Review and Meta-Analysis. Electronics (2023).) Research on training specific skills, particularly safety skills via VR, for individuals with ASD is scarce and lacks a systematic review. The purpose of this study is to determine whether virtual reality is equally effective in safety skills based on some existing relevant research. It is clear from the results of these studies that safety skills intervention via VR is equally effective. Secondly, only a small number of studies have focused on the relationship between the level of virtual reality immersion and intervention effects. We found no positive correlation between the two, that is, a higher level of virtual reality immersion does not necessarily result in better intervention effects. Certainly, this is a direction that could be pursued further. Thirdly, personalized VR intervention is also an emerging direction, which has only been added to driving skills training in current researches on safety skills. Personalized VR interventions depend on updates and developments in hardware and computer technology, such as improvements in sensor performance and machine learning algorithms. Therefore, personalized interventions for people with ASD via VR need to be developed in future studies. Later, we will continue this study by including more literature year by year.

Revised portion is marked red in Section 4 in the manuscript.

Comment 8:  The review could benefit from a more detailed analysis of the outcome measures used in the studies. Discussing the efficacy of VR interventions in a more nuanced way, considering different safety skills and individual differences, would be valuable.

Reply 8:  Thanks for your comments. We analyze the outcome measures used in the study in more detail in subsections 3.5 and 4.2.

Revised portion is marked red in the manuscript (Page 9, Line 321 - 331; Page 9, Line 309 - 319; Page 11, Line 427 - 432).

We tried our best to improve the manuscript and made some changes marked in red in revised paper which will not influence the content and framework of the paper. We appreciate for your warm work earnestly, and hope the correction will meet with approval. Once again, thank you very much for your comments and suggestions.

Reviewer 3 Report

Comments and Suggestions for Authors

The manuscript includes a narrative review on the use of virtual reality setups for helping people with autism in promoting safety skills. Twenty studies were reviewed and critically discussed. Results evidenced the effectiveness and suitability of VR-based interventions. The outcomes were adequately diversified and argued. I feel that relevant issues should be addressed according my comments. My pointes are listed below. 

1. In the Introduction a more comprehensive and solid theoretical framework should be provided on the use of VR in Autism and safety skills.- A strong rationale should be justified. 

2. I wonder whether both high and low functioning infdividuals with autism were considered. 

3. In the Inclusion and Ecluding criteria I could find only Inclusive criteria. Exclusion criteria, currently missing, should be added. 

4. I'm puzzled. Why at least three participants included in the studies were considered? Clarification is needed. 

5. The use of VR was considered on any targeted behaviors for autism or only on safety skills? Clarification is needed. 

6. Sub-headings 3.2 and 3.4 are both entitled Intervention subject. Is that a repetition? I would substitute subject with target or behavior or participant. 

7. A limitation/future research perspectives Section, currently missing, is warranted. 

Comments on the Quality of English Language

The English requires minor edits throughout. 

Author Response

Dear Reviewer:

Thank you very much for your comments and professional remarks concerning our manuscript entitled ‘Virtual Reality Utilized for Safety Skills Training for Autistic Individuals: A Review’ (ID: behavsci-2754222). These comments and suggestions are all valuable and very helpful for revising and improving our manuscript, and significantly important for our research. We have talked about these comments carefully and made modifications which we hope meet with your approval. We have provided a point-by-point response to your comments below.

Comment 1:  In the Introduction a more comprehensive and solid theoretical framework should be provided on the use of VR in Autism and safety skills.- A strong rationale should be justified.

Reply 1:  We have supplemented a theoretical framework for the application of VR to autism and safety skills in the introduction. The additions are listed below:

Virtual environments (VEs) are a compromise between artificial and natural environments with clear instructional goals and designs that meet the standards of teaching and learning environments for people with autism. They also offer controlled and individualized conditions that allow individuals with ASD to train in a controlled, repeatable environment without safety concerns [8]. Simultaneously, it offers diverse immersive experiences for users: including non-immersive, semi-immersive, and immersive [9]. Burdea and Coiffet summarize immersion, interactivity, and imagination as the basic characteristics of VR technology [10], also known as the 3I characteristics. Although the effectiveness of VR interventions has been measured and assessed in different ways across studies, in general, VR interventions are effective if the skills learned in the VR environment can be generalized to the natural environment. In addition, the use of VR in autism interventions has multiple theoretical supports: firstly, according to psychological theories of autism, the cognitive style of individuals with ASD is predominantly visually oriented [11]. And VR technology emphasizes exactly audio-visual responses. Notably, they prefer controlled interactions and able to respond well to challenges provided by computers with clear structure and consistent expectations. Moreover, learning theories such as task-based learning, adaptive learning, situated learning and simulation learning further enhance the theoretical support for the use of VR in ASD intervention, making its application even more convincing.

Revised portions are marked purple in the manuscript (Page 2, Line 59 - 77).

Comment 2:  I wonder whether both high and low functioning individuals with autism were considered. 

Reply 2:  Yes, both high and low functioning individuals with autism were considered.

Comment 3:  In the Inclusion and Excluding criteria I could find only Inclusive criteria. Exclusion criteria, currently missing, should be added. 

Reply 3:  Thanks for your comments. We have added exclusion criteria in sub-section 2.1.

Revised portions are marked purple in the manuscript (Page 3, Line 115 - 127).

Comment 4:  I'm puzzled. Why at least three participants included in the studies were considered? Clarification is needed. 

Reply 4:  The studies we include must be empirical and therefore must have at least one experimental subject, the modifications we have made are as follows:

The study should include at least one patients diagnosed by a clinician or identified as having ASD using a standardized diagnostic tool.

Revised portions are marked purple in the manuscript (Page 3, Line 117).

Comment 5:  The use of VR was considered on any targeted behaviors for autism or only on safety skills? Clarification is needed.

Reply 5:  Thanks for your comments. Virtual reality can be used on any of the target behaviors in autism, as noted in the fourth paragraph of the introduction. But in this paper, we only discuss and analyze the use of virtual reality for safety skills in autism.

The exact description is marked purple in the manuscript (Page 2, Line 84 - 97).

Comment 6:  Sub-headings 3.2 and 3.4 are both entitled Intervention subject. Is that a repetition? I would substitute subject with target or behavior or participant. 

Reply 6:  Thanks for your comments. We are sorry for our carelessness. We have modified Sub-heading 3.4 to Safety skill generalization.

Revised portion is marked purple in the manuscript (Page 9, Line 320).

Comment 7:  A limitation/future research perspectives Section, currently missing, is warranted. 

Reply 7: Thanks for your suggestion, we have added this section to the article. The additions are as follows:

4.4. Limitations and future directions

In order to improve the independence and safety of individuals with ASD and to provide some guidance for future safety skills training, this study reviewed and analyzed relevant literature of the past 10 years (2014-2023). Later, we will continue this study by including more literature year by year. Of course, this study is not without its limitations: the exclusion of published studies other than English language ones was among the limitations and challenges of this study. In addition, in terms of intervention effects, this paper focuses on the influence of the degree of immersion in VR on it, and in fact, the subjects' age, intelligence, etc. may also be some of the influencing factors, which should be explored in future research.

Added portion is marked purple in the manuscript (Page 14, Line 553 - 562).

We tried our best to improve the manuscript and made some changes marked in purple in revised paper which will not influence the content and framework of the paper. We appreciate for your warm work earnestly, and hope the correction will meet with approval. Once again, thank you very much for your comments and suggestions.

Round 2

Reviewer 1 Report

Comments and Suggestions for Authors

This version of the article provides more explanations and is more satisfying in terms of clarifying concepts and arguments. These efforts on the part of the authors are much appreciated. One or two minor comments have been included in this version of the article to further improve its quality.

Reviewer 2 Report

Comments and Suggestions for Authors

Accept.

Author Response

Dear reviewer:

Thank you very much for the strong support to our work.

Reviewer 3 Report

Comments and Suggestions for Authors

The paper can be accepted in its current form. 

Author Response

(The authors gave the same response as above.)
